



# Contrasting regional variability of buried meltwater extent over two years across the Greenland Ice Sheet

Devon Dunmire[1], Alison F. Banwell[2], Jan T. M. Lenaerts[1], and Rajashree Tri Datta[1]

[1]Department of Atmospheric and Oceanic Sciences, University of Colorado Boulder, USA
[2]Cooperative Institute for Research in Environmental Sciences (CIRES), University of Colorado Boulder, USA

**Correspondence:** Devon Dunmire (devon.dunmire@colorado.edu)

**Abstract.** The Greenland Ice Sheet (GrIS) rapid mass loss is primarily driven by an increase in meltwater runoff, which highlights the importance of understanding the formation, evolution and impact of meltwater features on the ice sheet. Buried lakes are meltwater features that contain liquid water and exist under layers of snow, firn, and/or ice. These lakes are invisible in optical imagery, challenging the analysis of their evolution and implication for larger GrIS dynamics and mass change. Here,

we present a method that uses a convolutional neural network, a deep learning method, to automatically detect buried lakes across the GrIS. For the years 2018 and 2019, we compare total areal extent of both buried and surface lakes across six regions, and use a regional climate model to explain the spatial and temporal differences. We find that the total buried lake extent after the 2019 melt season is 56% larger than after the 2018 melt season across the entire ice sheet. Northern Greenland observes the largest increase in buried lake extent after the 2019 melt season, which we attribute to late-summer surface melt and high

autumn temperatures. We also provide evidence that different processes are responsible for buried lake formation in different regions of the ice sheet. For example, in western Greenland, buried lakes often appear on the surface during the previous melt season, indicating that these features form when surface lakes partially freeze and become insulated as snowfall buries them. In contrast, in southeast Greenland, most buried lakes never appear on the surface, signifying that these features may form due to subsurface penetration of shortwave radiation and/or downward percolation of meltwater. This study helps to provide

additional perspective on the potential role of meltwater on GrIS dynamics and mass loss.

## 1 Introduction

The Greenland Ice Sheet (GrIS), which holds enough ice to raise sea level globally by more than 7 m (Church and Gregory, 2001; Smith et al., 2020) has experienced net mass loss every year since 1998 (Mouginot et al., 2019). Since 1972, the GrIS has contributed a total of 13.7 ± 1.1 mm of sea level rise. Prior to 2005, ice discharge was the primary driver of Greenland

mass loss (Enderlin and others, 2014). However, meltwater runoff, which has accelerated recently, is now the dominant factor in Greenland mass loss (Smith et al., 2020; Van Den Broeke et al., 2016; Enderlin and others, 2014). Thus, surface melt plays an increasingly important role in the mass balance of the GrIS.

Melting is exacerbated by the positive melt/albedo feedback, whereby melting acts to lower surface albedo, which in return allows for greater absorption of incoming solar radiation, thus further enhancing surface melt (Box et al., 2012; Lüthje et al.,



2006; Tedesco et al., 2012). Meltwater often pools in surface lakes in the ablation zone during the summer from May to October (McMillan et al., 2007; Banwell et al., 2012). Water in these lakes then either runs off the ice sheet, drains via hydrofracture (Das et al., 2008; Tedesco et al., 2013; Williamson et al., 2018b), or refreezes in the firn (Bell et al., 2018). Firn acts as a sponge that takes up meltwater and buffers against mass loss (Harper et al., 2012). Water that refreezes in near-surface firn leads to the formation of ice lenses, which increases the density and decreases the porosity of the near-surface firn, thus reducing the

capacity of Greenland's firn to hold future meltwater (Machguth et al., 2016). Ice lenses due to refrozen meltwater are rapidly increasing in areal extent across the GrIS, leading to increased runoff (MacFerrin et al., 2019).

However, not all meltwater stored within the firn refreezes. In fact, some meltwater remains liquid, buried several meters below the surface (Koenig et al., 2015). These shallow buried lakes (or subsurface lakes) have been discovered later in the summer and at higher elevations than their surface counterparts (Miles et al., 2017; Lampkin et al., 2020). Liquid water exists

in buried lakes even throughout the winter (Schröder et al., 2020) and some have been shown to rapidly drain, delivering liquid water to the bedrock during the winter months (Benedek and Willis, 2020).

Because meltwater runoff is the dominant driver of mass loss from the GrIS, developing methods to detect surface water has been the focus of many recent studies (Yuan et al., 2020; Sundal and others, 2009; Williamson et al., 2017; Liang et al., 2012). Many of these methods, however, rely on optical imagery to detect meltwater. Buried lakes, in contrast, are invisible

in optical images, challenging the study of their extent, evolution, and interaction with drainage systems using conventional methods. More recently, studies have begun using Synthetic Aperture Radar (SAR) to detect both surface and buried lakes (Miles et al., 2017; Johansson and Brown, 2012; Dunmire et al., 2020; Schröder et al., 2020; Benedek and Willis, 2020). As an active sensor, SAR does not require a light source and thus is useful at night or during the polar winter. The European Space Agency's Sentinel-1 satellite uses C-band radiation, which is particularly useful because it is capable of penetrating clouds,

ice and snow up to several meters (Rignot et al., 2001). Water is a strong absorber of C-band SAR radiation. Because of this, Sentinel-1 microwave backscatter can be used as a diagnostic indicator of both the presence of melt and melt intensity, and can even detect liquid meltwater buried beneath several meters of ice and snow (Miles et al., 2017).

Fast, automatic detection of buried water features across large spatial and temporal scales is an important step in better understanding their impact on GrIS mass balance. In this study we develop a convolutional neural network (CNN), a deep

learning technique for automatic detection of features from images, to detect buried lakes across the GrIS. CNNs are beneficial for feature detection because of their ability to learn spatial relationships from two-dimensional images, and because they can easily be applied across large spatial and temporal scales. CNNs are becoming an increasingly popular choice for automatic feature detection in polar regions and have been used for detecting features such as glacier calving margins (Mohajerani et al., 2019; Zhang et al., 2019), icebergs (Rezvanbehbahani et al., 2019), sea ice concentration (Wang et al., 2016, 2017; Cooke

and Scott, 2019; Song et al., 2019), and surface water (Daneshgar et al., 2019; Yuan et al., 2020). Here, we develop and use a CNN to compare the distribution of buried lakes in Greenland's six subregions (Rignot and Mouginot, 2012) during a relatively cold, low-melt year (2018) and a warmer, high-melt year (2019). We compare both buried lake distribution and surface lake distribution between the two years, and subsequently use a regional climate model to help explain the spatial and temporal





differences. Finally, we investigate additional imagery prior to buried lake detection to hypothesize that different processes are
responsible for the formation of these features in different regions of the GrIS.

## 2 Methods

### 2.1 Buried lake detection

#### 2.1.1 Training/testing dataset preparation

Our study region covers the entire GrIS. To create training and testing data sets we collected Sentinel-2 (S2) optical imagery
and Sentinel-1 (S1) C-band SAR microwave backscatter imagery from all six GrIS subregions. S2 images were collected
during the late summer (September) of 2016 and 2017 and S1 images were collected during the winter between January 1 and
January 7 following the melt season to minimize firn saturation than can obscure buried lakes in microwave imagery. Google
Earth Engine (Gorelick et al., 2017) (GEE) was used to collect all images used in this study. GEE preprocesses S1 images
with the following steps: 1) thermal noise removal, 2) radiometric calibration, 3) terrain correction using ASTER DEM, and
4) values converted to decibels via log scaling. S2 images are available as top of atmosphere reflectance divided by 10000. All
S1 and S2 images exported from GEE were rescaled to 30x30 m resolution (to increase processing speed) on the same grid
using nearest neighbor resampling and reprojected to an Arctic stereographic grid (EPSG:3995). Images were broken up into
256x256 and 512x512 pixel tiles, all resized to 256x256 pixels using a first order spline interpolation. For each tile, a false
colour image was created using both the S1 and S2 imagery. The false colour images are made up of the following bands:

– Band 1 = Greyscale (from S2) = 0.299 * Red + 0.5870 * Green + 0.1140 * Blue

   – Band 2 = HV band (from S1) normalized from 0 to 1 with respect to the individual image tile such that the minimum tile
     pixel is 0 and the maximum tile pixel is 1. The purpose of this band is to emphasize microwave backscatter changes that
     result from the presence of liquid water.

   – Band 3 = HV band (from S1) normalized between -30 and 0. This band functions to mute backscatter changes that are
small with respect to the surrounding area, and thus not likely due to the presence of buried liquid water.

Image tiles were manually classified into seven different categories 1) uniform ice and snow (Fig. 1a, 1277 tiles total), 2)
textured ice and snow (Fig. 1b, 1362 tiles total), 3) surface lake remnants (Fig. 1c, 1322 tiles total), 4) open ocean and sea water
(Fig. 1d, 99 tiles total), 5) buried lakes (Fig. 1e, 1399 tiles total), 6) land (Fig. 1f, 542 tiles total), and 7) mountains surrounded
by snow (Fig. 1g, 227 tiles total). 70% of the image tiles for each class were used for model training, 15% for model validation
during the training process, and 15% for a final model test. The training data set was augmented such that at each iteration of
model training, image tiles were randomly flipped horizontally and/or rotated by a random increment of 30°.




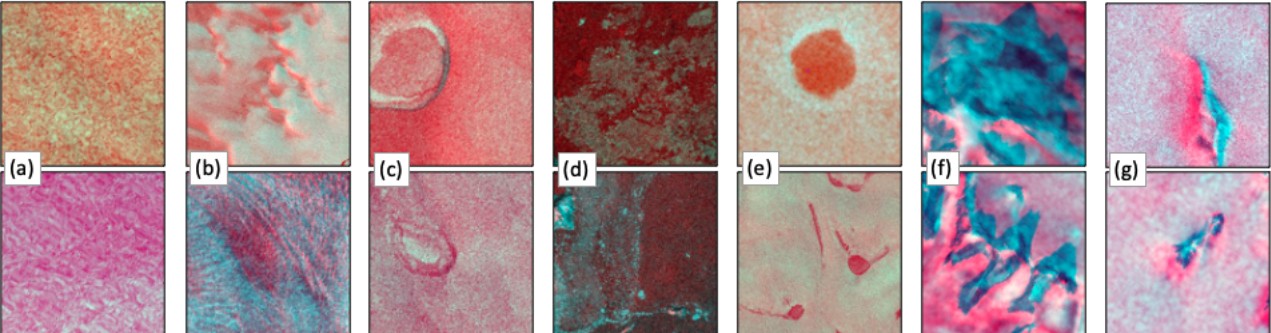

**Figure 1.** Two example false colour image tiles for each class used for CNN training. **(a)** Class 1: uniform ice and snow. **(b)** Class 2: textured ice and snow including bare ice regions (above) and crevassed regions (below). **(c)** Class 3: surface lake remnants. **(d)** Class 4: open ocean and sea water. **(e)** Class 5: buried lakes. **(f)** Class 6: land. **(g)** Class 7: mountains surrounded by snow.

### 2.1.2 Model training

A CNN was the method of choice for detecting subsurface lakes across Greenland. Unsupervised learning has been shown to be unsuitable for similar analysis on Antarctica (Moussavi et al., 2020) and CNNs have proven superior over other supervised classification algorithms at detecting surface water on Greenland (Yuan et al., 2020). CNNs are being increasingly used for land and surface classification problems (Maggiori et al., 2017). Here, we use the pre-trained AlexNet architecture (Krizhevsky et al., 2012) because it outperformed other model architectures on our validation dataset (Appendix Table A1). To optimize the model for our specific goal of detecting buried lakes, the fully connected layer was rebuilt using one which we designed. This layer used the Rectified Linear Unit (ReLU) activation function (Hara et al., 2015) to solve the vanishing gradient problem and implemented a dropout layer to prevent the model from overfitting (Srivastava et al., 2014). To optimize model parameters, we used the negative likelihood log loss function and Adam optimizer (Kingma and Ba, 2015). The learning rate began as 0.005, decreasing by a factor of 0.1 every 7 epochs using a scheduler. Our model was trained for 20 epochs with a batch size of 32 false colour training image tiles. Finally, our model was further optimized by feeding incorrectly classified tiles back into the training data set.

### 2.1.3 Model evaluation and testing

Two different methods were used to evaluate model performance: F1 score, and receiver operating characteristic (ROC) curves. F1 score (Eq. 1) is the harmonic mean of precision and recall. Precision (Eq. 2) is essentially a measure of how many positive classifications are actually buried lakes, while recall (Eq. 3) is a measure of how many buried lakes are correctly classified.

$$F1 = \frac{precision * recall}{precision + recall} \tag{1}$$



$$Precision = \frac{TP}{TP + FP} \tag{2}$$


$$Recall = \frac{TP}{TP + FN} \tag{3}$$

*TP = true positive, FP = false positive, FN = false negative*

Our second model performance metric is a ROC curve. This metric compares true and false positive rates at different classification thresholds. For example, with a threshold of 0, any image tile that the model says has >0% chance of being a

buried lake, is classified as such. Thus, at a threshold of 0, all image tiles will be classified as buried lakes and there will be high true and false positive rates. Similarly, for a threshold of 1, no image tiles will be classified as buried lakes and there will be low true and false positive rates. The area under the ROC curve (AUC) is used as an evaluation of model performance, with larger AUC corresponding to better models.

Appendix Table A1 compares F1 score and AUC for all model architectures we trained to detect buried lakes. Our initial

iteration of AlexNet had a relatively high recall and low precision. However, we decided that it was more important to mitigate false positives than to detect all possible subsurface lakes, thus prioritizing precision over recall. Therefore, to minimize false positives, we chose a relatively high threshold of 0.7 for classifying buried lakes. Using this threshold, our final model had a buried lake precision of 0.929, recall of 0.880, and F1 score of 0.904. Precision for all classes can be seen in Appendix Figure A1, which illustrates the confusion matrix for the test data set.

**2.1.4 Classification**

To detect buried lakes across the GrIS, we followed the workflow outlined in Figure 2. First, we collected S1 and S2 images across the entire periphery of the GrIS for the years 2018 and 2019, and followed the same data collection and pre-processing steps as outlined above in Sect. 2.1.1. As described above, we used a threshold of 0.7 for buried lake classification such that if an image tile had greater than a 70% probability of being a buried lake, according to the model, the tile was classified as such.

We also created a mountain/land mask to ensure that mountain areas were not erroneously classified as buried lakes. The mask was created by masking out areas where the grayscale band was <0.4 in image tiles that had >75% of being either a mountain or land tile, according to the model. The mask was dilated by 20 pixels to include regions of the ice that could have been affected by mountain shadows.

We then used a combination of thresholding and morphological operations to outline individual buried lakes in the image

tiles. First, we applied a threshold to band 1 of the false colour image (normalized HV backscatter from S1) such that all but the darkest 5% of pixels were masked out. We next performed a series of opening/closing operations using 3x3, 6x6, and finally 12x12 pixel kernels to fill holes in the lake and mask out darker pixels outside the lake. All binary image tiles were then mosaicked and the individual buried lakes were contoured. We found that repeating this process of thresholding and using morphological operations improved the delineation of buried lakes so these steps was repeated for each contoured area.



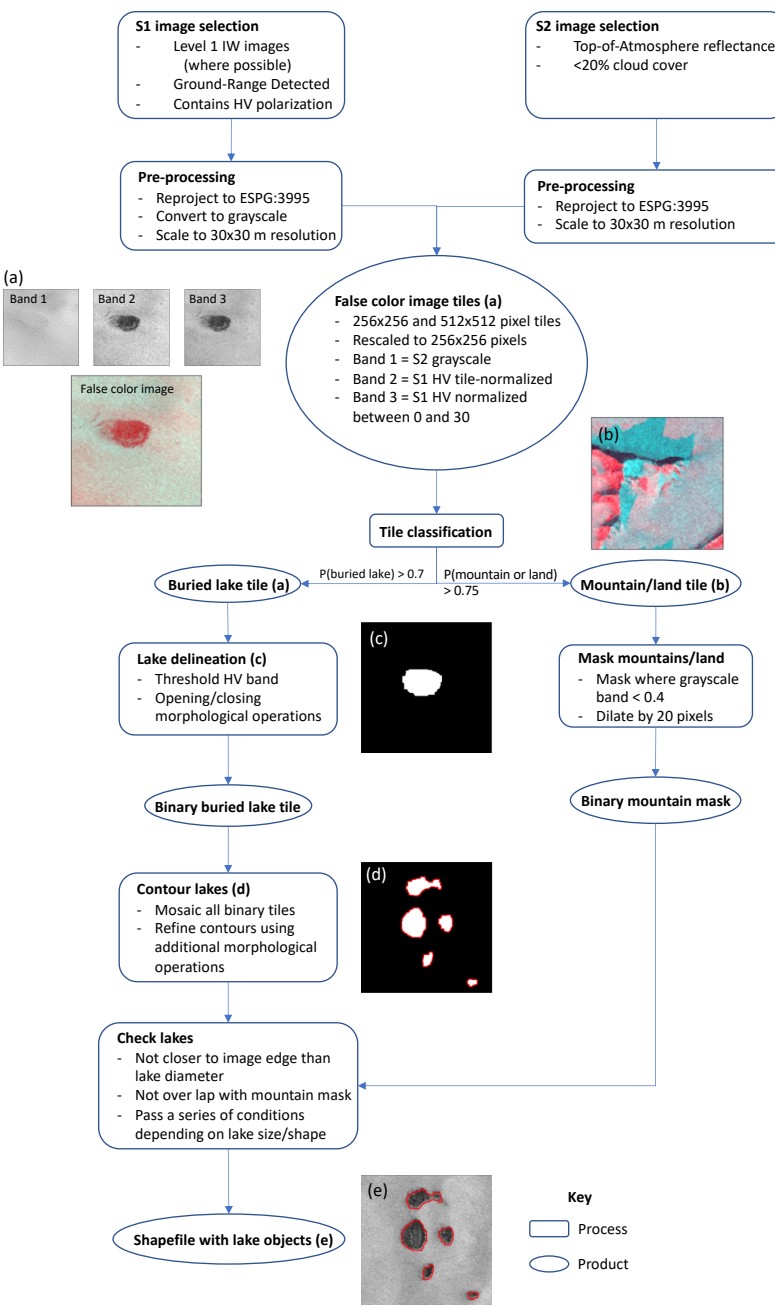

**Figure 2.** Flow diagram of the methodology with key steps illustrated in image panels (a-e). The final product is shown in (e).





Finally, each contoured area had to meet a series of conditions to be considered a buried lake (Appendix Fig. A2). These conditions, which depend on the shape and size of the lake, were optimized by manual inspection to avoid false positives. For example, for smaller buried lakes to be considered, they had to have a lower (compared to larger lakes) microwave backscatter compared to the surrounding background area.

## 2.2    Surface lake detection

We used all S2 images with <10% cloud cover, rescaled from the native resolution of 10x10 m to 30x30 m resolution (to match S1 image resolution for CNN work and for increased processing speed), from GEE to detect surface lakes across the GrIS during the 2018 and 2019 melt seasons. We followed Williamson et al. (2018a) to mask out clouds by removing pixels in which the band 11 (SWIR) top-of-atmosphere (TOA) reflectance exceeded a threshold of 0.140. To mask ice-marginal areas, we followed Moussavi et al. (2020) by removing pixels with a Normalized Difference Snow Index (NDSI = $\frac{Green-SWIR}{Green+SWIR}$) <

0.85 and where band 2 (blue) < 0.4. Pixels above a Normalized Difference Water Index (NDWI = $\frac{Blue-Red}{Blue+Red}$) threshold of 0.5 (Miles et al., 2017) were considered to be a surface lake pixel. This threshold is higher than that used in other studies (Yang and Smith, 2013; Williamson et al., 2018a; Benedek and Willis, 2020), but we found that using a lower threshold resulted in the erroneous inclusion of cloud and mountain shadows in the analysis. Additionally, use of this higher threshold was appropriate for our goal of comparing relative surface water ponding between the 2018 and 2019 melt seasons in each GrIS subregion. For

lakes with small areas of floating surface ice, we filled in these areas by twice performing a morphological closing operation using a 5x5 pixel kernel. We only considered surface lakes that have an area > 0.05 km$^2$, which is consistent with our area threshold for buried lake detection.

## 2.3    Buried lake analysis

We compared buried and surface lake elevations by calculated the average elevation of each lake using the Greenland Ice
Mapping Project (GIMP) elevation dataset (Howat et al., 2015). GIMP estimates of surface elevation have an ice-sheet-wide root-mean-square error relative to ICESat elevations of ± 10 m above sea level (asl) ranging from a minimum of ± 1 m asl in most ice areas to ± 30 m asl in high relief regions (Howat et al., 2015).

To investigate the buried lake formation process in different regions of the GrIS, we determined the percentage of detected buried lakes in each region that showed any evidence of surface meltwater during the previous summer. We define "evidence of
surface meltwater" as the presence of > 5 pixels with an NDWI > 0.25 within the bounds of the buried lake. For several buried lakes, we also examined a combination of optical imagery from Sentinel-2, Landsat 8 and Landsat 7 from summers dating back to 2012. We also looked at surface topography profiles above select buried lakes using the Arctic DEM Mosaic (Porter et al., 2018) at 2 m resolution.

Finally, we compared buried lake distribution with the locations of firn aquifers from 2015-2017 using the Miège (2018) firn
aquifer dataset derived from Operation Ice Bridge radar.



## 2.4 Regional climate model analysis

To relate our buried lake detection results in the two consecutive years to the interannual variations in surface climate and surface mass balance, we used output from the regional climate model RACMO2.3p2 (Noël et al. (2018); RACMO2 hereafter). RACMO2 provides output of Greenland climate and surface mass balance from 1958 to 2020 at 5.5 km horizontal resolution, which is then further statistically downscaled to 1 km horizontal resolution (Noël et al., 2016). We refer to Noël et al. (2018) for a detailed description and evaluation of RACMO2 over the GrIS.

Here, we used RACMO2-derived monthly mean fields of Greenland temperature, surface melt, precipitation, and surface mass balance from 1 June to 31 December in 2018 and 2019, and compare these to the long-term 1958-2017 climatology.

## 3 Results

### 3.1 Surface and buried lake detection

We detect a total of 374 buried lakes (covering a total area of 241 km$^2$) following the 2018 melt season (Fig. 3a, Appendix Table B1) and a total of 599 buried lakes (covering 376 km$^2$) following the 2019 melt season (Fig. 3b, Appendix Table B1). In both years, buried lakes are concentrated along the western coast in the SW, CW, and NW regions of the GrIS. Detected buried lakes in SW Greenland have the largest average lake area of all regions (0.77 km$^2$ following the 2018 melt season, and 0.90 km$^2$ following the 2019 melt season), and include the single largest buried lake detected with an area of 3.95 km$^2$ in 2019. Buried lakes range in elevation from approximately 455 to 2450 m asl (Appendix Fig. B2), with the highest buried lakes located in the SW and SE (average elevations of 1831 and 1710 m asl, respectively) and the lowest buried lakes located in the NE, NO, and NW (average elevations of 1195 m asl, 1135 m asl, and 1144 m asl, respectively). These results are broadly consistent with Miles et al. (2017) and Koenig et al. (2015), who found that the majority of buried lakes ranged in elevation from 1000 to 2000 m asl. In general, buried lakes do not appear at different elevations in 2019 than in 2018, with the exception of the NO and NE regions. However, these exceptions may be the result of bias due to a small number of buried lakes detected in 2018 in these regions.

We also find that surface lake extent is much lower during the 2018 melt season than during the 2019 melt season. Using our method to detect surface lakes (NDWI > 0.5, and area > 0.05 km$^2$), we detect 3846 surface lakes (covering 1242 km$^2$) in 2018 (Appendix Fig. B1, Appendix Table B2) and 6146 surface lakes (covering 2569 km$^2$) in 2019 (Appendix Fig. B1, Appendix Table B2). Also similar to buried lake distribution, the average elevation of detected surface lakes is highest in the SW and SE regions (average 1365 m asl and 1335 m asl, respectively) and lowest in the NO, NW, and NE regions (average 718 m asl, 860 m asl, and 962 m asl, respectively). In all regions for both years, the average elevation of detected buried lakes is higher than for detected surface lakes (Appendix Fig. B2), which is consistent with previous work (Miles et al., 2017; Lampkin et al., 2020).

On an ice sheet wide scale, both the total surface and buried lake extents are much greater in 2019 than in 2018. Regionally, however, different patterns emerge between surface and buried lake distribution for the two years. Figure 4a shows that the



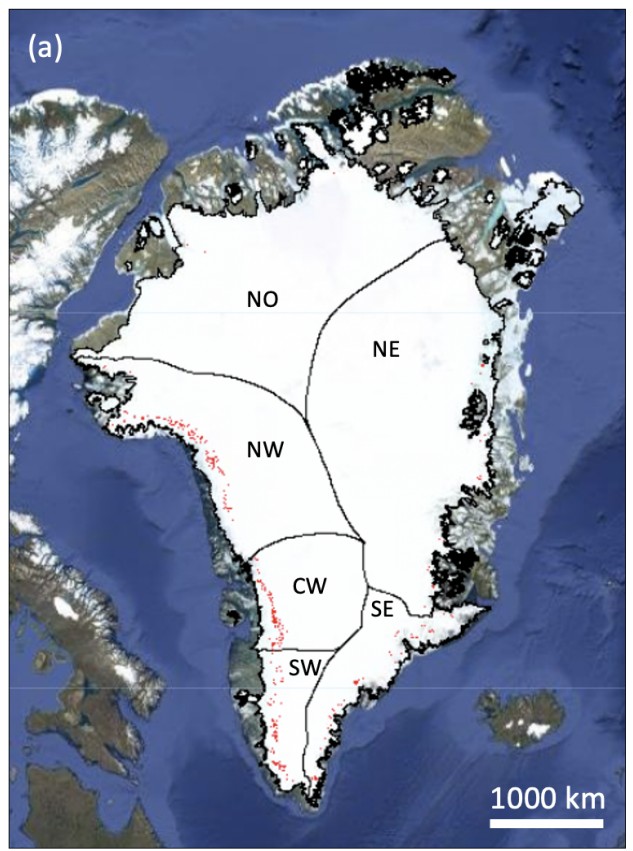
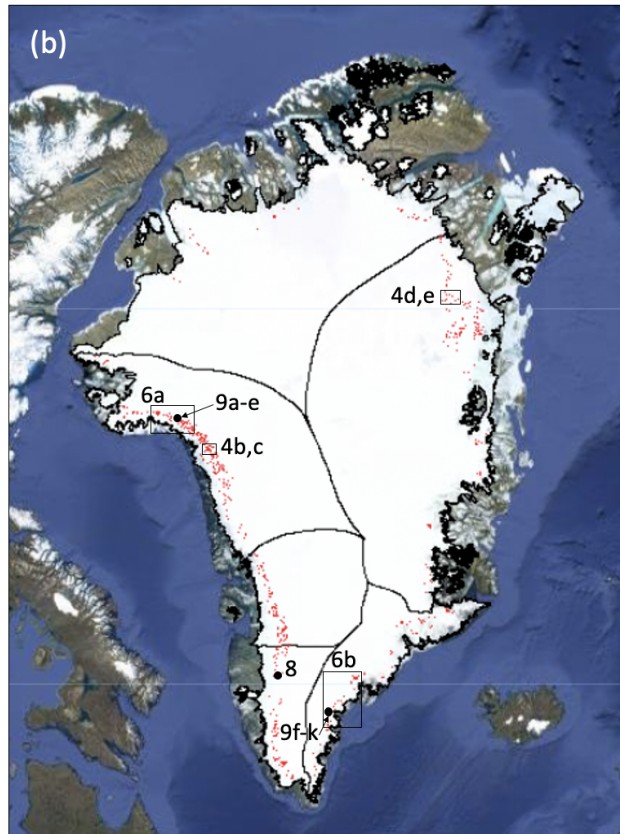

**Figure 3.** Buried lake distributions, shown in red. (a) 2018 buried lakes with major GrIS drainage basins labeled (Rignot and Mouginot, 2012). (b) 2019 buried lakes with the area used in subsequent figures outlined by black boxes and labeled. Background map is from GEE Gorelick et al. (2017).

ratio of surface lake area in 2018 to 2019 is much less than 1 across all GrIS subregions, ranging from 0.32 in NE Greenland to 0.67 in SE Greenland. The 2018:2019 buried lake area ratio, however, varies much more across the six subregions of the GrIS (Fig. 4a). In the SW and CW regions, we find approximately the same 2018:2019 buried lake area ratio (0.931 and 0.996, respectively), even though surface lake extent is much less in 2018. In contrast, in the NO and NE regions, the 2018 to 2019 buried lake area ratio is even smaller than the surface lake area ratio (2018:2019 buried lake ratios of 0.133 and 0.171, respectively). The NW and SE regions have comparable 2018:2019 surface and buried lake area ratios. Figure 4b-e shows detected buried lakes in NW (b,c) and NE (d,e) Greenland for 2018 and 2019, highlighting the 2019 increase in buried lake area in these regions.

For each buried lake, we additionally analyzed select S2 optical imagery from the previous melt season for any evidence of surface water (see Sect. 2.3). Figure 5 shows how the percentage of buried lakes with evidence of liquid water on the surface


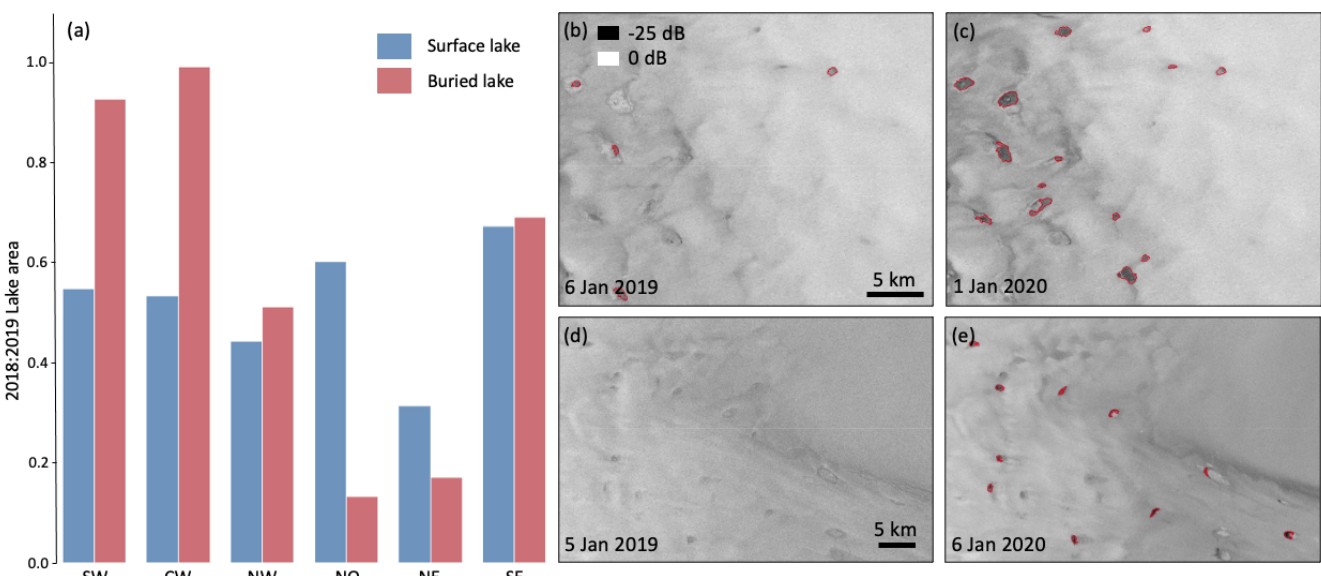

**Figure 4.** Comparison of 2018 and 2019 total surface and buried lake areas. (a) Ratio of 2018 to 2019 total detected surface (blue) and buried (red) lake area for each major GrIS drainage basin. (b-e) S1 images from NW Greenland (b,c) and NE Greenland (d,e), with detected subsurface lakes outlined in red. (b) 6 Jan 2019 (following the 2018 melt season). (c) 1 Jan 2020 (following the 2019 melt season). (d) 5 Jan 2019. (e) 6 Jan 2020.

during the previous melt season varies across the six GrIS subregions. In SW Greenland, most of the detected buried lakes have some surface meltwater within their bounds during the previous melt season (92% in 2018 and 85% in 2019). In contrast, very few lakes in the NW (5.9% in 2018 and 5.8% in 2019) and SE (1.8% in 2018 and 4.2% in 2019) have any evidence of previous summer surface melt within their bounds. This discrepancy indicates that buried lakes may form via different processes in different regions of the GrIS and is something we address in the Sect. 4.3.

Buried lakes in NW and SE Greenland also coincide with the locations of, and could be connected to, firn aquifers that have been detected in these regions (Fig. 6, Koenig et al. (2014); Forster et al. (2014); Miège et al. (2016); Brangers et al. (2020)). It makes sense that these features are co-located because they both require similar climatic conditions of high melt and high accumulation to exist (Koenig et al., 2014; Munneke et al., 2014; Dunmire et al., 2020). The buried lakes detected in this study are likely located at shallower depths than the firn aquifers because they can be detected in S1 imagery, unlike most firn aquifers which are too deep to be detected directly in S1 imagery. It is unclear what relationship the buried lakes and firn aquifers have, but it is possible that the buried lakes feed the firn aquifers by draining vertically.

## 3.2 Regional climate model analysis

To investigate the discrepancies between total surface and buried lake area across the six GrIS subregions, we analyzed RACMO2 data from 2018 and 2019 at elevations lower than 2500 m, comparing temperature and melt with the climatological



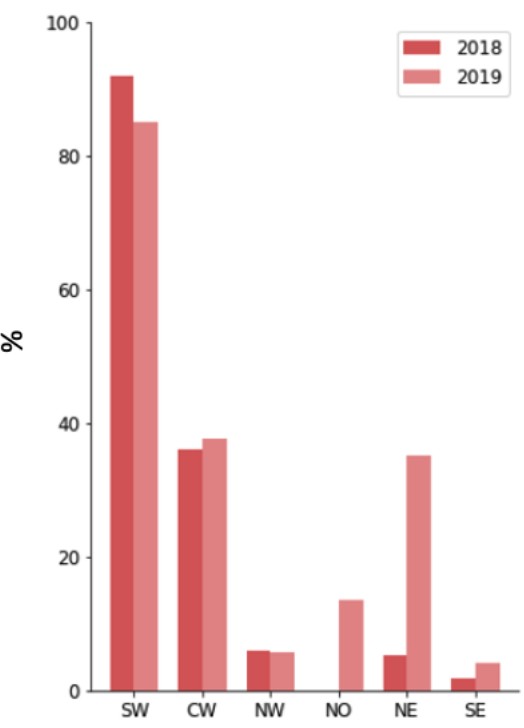

**Figure 5.** Percentage of buried lakes that show any evidence of meltwater on the surface (> 5 pixels with an NDWI > 0.25) during the previous melt season, for each of the six subregions.

mean (Appendix Fig. B3, B4). We find no spatial or temporal patterns associated with monthly precipitation or surface mass balance anomalies that may explain surface and buried lake differences so this analysis is not included here. Compared to 2018, 225  2019 is much warmer across elevations lower than 2500 m in every month from June to November (Fig. 7a). During June and July 2019, high monthly mean temperatures (Fig. 7a) are present across the entire ice sheet (+1.52°C for June and +1.62°C for July). It is very likely that these anomalously high temperatures contribute to anomalously high melt during these months (Fig. 7b, 1.51 standard deviations higher melt than the climatological mean for June and 1.02 standard deviations higher melt than the climatological mean for July). In contrast, in 2018, June and July saw average temperature anomalies of −0.17°C and 230  −1.00°C, respectively and lower melt (0.24 standard deviations less than the climatological mean for June and 0.33 standard deviations less than the climatological mean for July). These ice-sheet-wide climatological conditions can explain the greater total surface lake area detected in 2019 compared to 2018 in all six GrIS subregions. However, from August to November 2019, a new pattern emerges over northern Greenland, marked by anomalously high temperatures and melt. In the NW, NO and NE regions, the 2019 temperature anomalies are +1.17 °C, +1.38 °C, +1.96 °C, and +2.00 °C for August, September, 235  October, and November, respectively. The August 2019 melt is 2.26 standard deviations higher than the climatological mean (compared to only 0.26 standard deviations higher in August 2018). In comparison, in the CW, SW and SE regions, the 2019





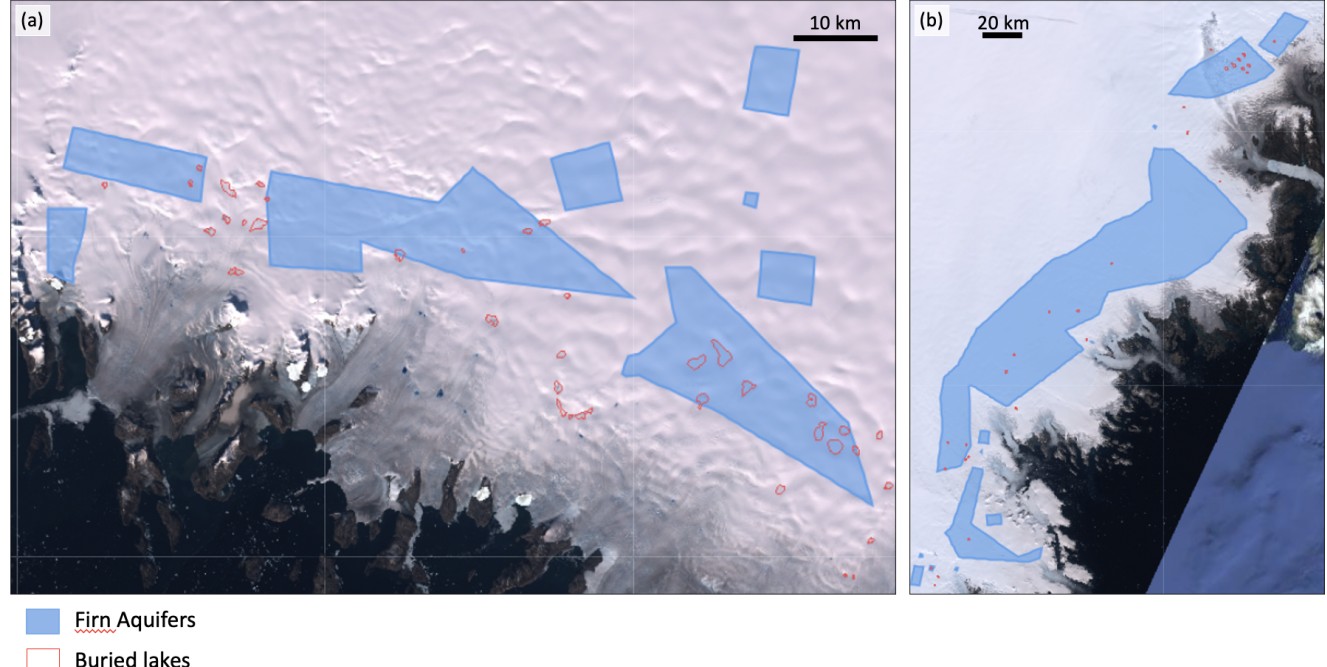

**Firn Aquifers**

**Buried lakes**

**Figure 6.** Firn aquifer and detected buried lake locations. (a) 2015-2017 NW GrIS firn aquifers from Operation Ice Bridge (OIB) (Miège, 2018) and buried lakes following the 2019 melt season. Background image is a S2 image from 1 September 2019. (b) 2015-2017 SE GrIS firn aquifers from OIB and buried lakes following the 2019 melt season. Background image is a S2 image from 21 July 2019.

temperature anomalies for these same months are much less remarkable at $-0.20\,°C$, $+0.26\,°C$, $+0.51\,°C$, and $+1.60°C$ and the August 2019 melt anomaly was only 0.52 standard deviations higher than climatological mean. These results suggest for the three regions in northern Greenland (NW, NO and NE), higher melt in the late summer of 2019, coupled with anomalously

high autumn temperatures, may contribute to the higher total buried lake area detected here in 2019 compared to elsewhere on the GrIS.

## 4 Discussion

### 4.1 Buried lake detection limitations

The number of buried lakes detected in this study is likely an underestimation of the actual number of buried lakes that

exist across the GrIS in 2018 and 2019, for several reasons. Firstly, our detection of buried lakes is dependent on prescribed thresholds based on the microwave backscatter within the buried lake bounds relative to the surrounding area (Appendix Fig. A2). We set these thresholds in a conservative manner that prioritizes minimizing false positives, but as a result, our analysis likely misses some buried lakes that do not meet our thresholds. For example, if we increase the thresholds we use in this study





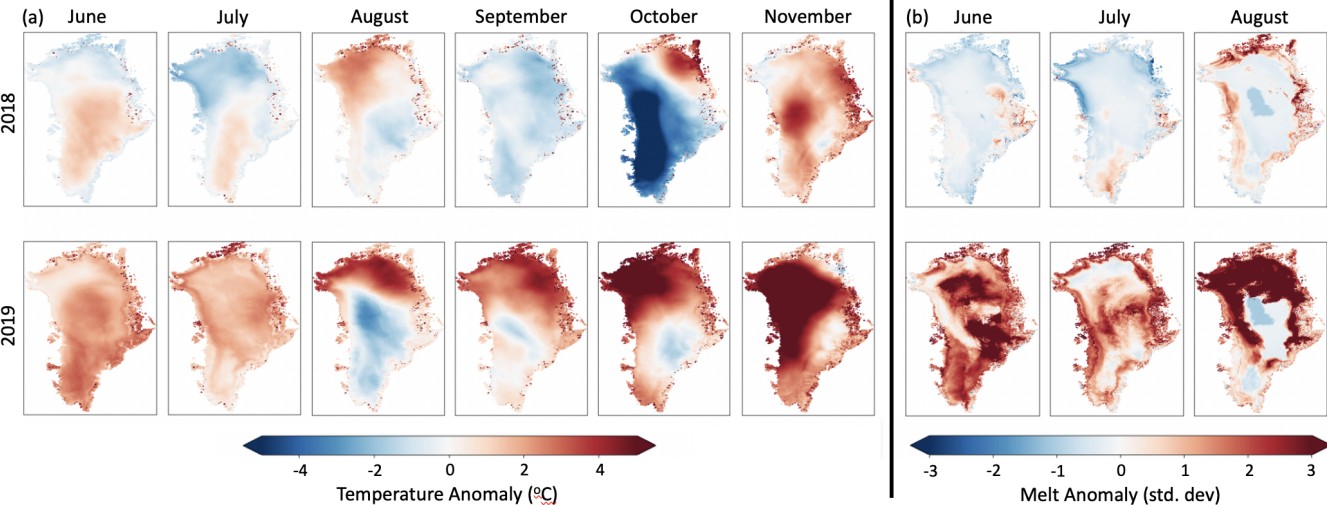

**Figure 7.** RACMO2 2018 and 2019 climatological anomalies. (a) June through November monthly temperature anomaly (°C) for 2018 (upper) and 2019 (lower) from the 1958-2017 climatological mean. (b) June through August monthly melt standard deviation for 2018 (upper) and 2019 (lower) from the 1958-2017 climatological mean.

by 5% (which makes the result less conservative), we see 5.2% and 5.0% increases in the total detected buried lake volume in
2018 and 2019, respectively. However, this changes the 2018:2019 buried lake area ratio by only +0.18%. Conversely, a 5%
decrease in the thresholds we use in this study (which makes the results even more conservative) results in 11.0% and 5.4%
decreases in the total detected buried lake volume in 2018 and 2019, respectively. This decreases the 2018:2019 buried lake
area ratio by 5.8%, indicating that the difference in backscatter between a buried lake and its surroundings was greater in 2018
than it was in 2019. A possible explanation for this is that the buried lakes detected in 2018 had persisted from the 2017 melt
season or before, resulting in these lakes being buried at greater depths than the lakes that formed after the 2019 melt season.
Another possible explanation is that buried lakes contained less water in 2018 than in 2019, making the lakes in 2018 appear
less different relative to their surrounding regions in the S1 images.

Another reason why our method may underestimate the number of buried lakes is because we do not consider lakes that are
smaller than 0.05 km$^2$ (approximately 56 pixels), so there may be smaller buried lakes that were not included in the analysis.
Additionally, as the penetration depth of C-band microwave radar in ice and snow is several meters (Rignot et al., 2001), buried
lakes located at depths greater than this will not be detected using our method. Finally, recent work has shown that buried lakes
can sometimes drain during the winter months Benedek and Willis (2020). Our analysis does not include buried lakes that may
have already drained during the winter before January 1; the first date from which we acquire S1 imagery to detect buried lakes
following the previous melt season.





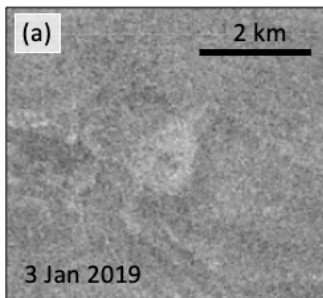 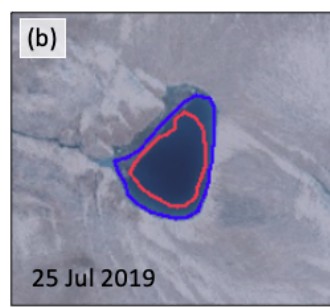 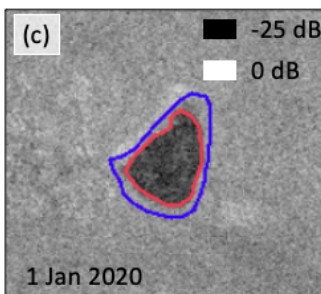

**Figure 8.** Chronological S1 (greyscale) and S2 (true colour) images of a buried lake detected in SW Greenland in the winter following the 2019 melt season. (a) S1 image from 3 January 2019 with no buried lake detected. (b) S2 image from 25 July 2019 with surface lake extent outlined in blue and post-melt season buried lake extent outlined in red. (c) S1 image from 1 January 2020.

## 4.2 Buried lake formation processes

Previous studies have generally assumed that buried lakes on the GrIS develop when a surface lake partially freezes through and then gets buried by snowfall, insulating any remaining liquid water under the surface (Koenig et al., 2015; Schröder et al., 2020). Modelling efforts have confirmed that this process is sufficient to allow liquid water to exist under the ice surface throughout the winter on both the Greenland and Antarctic ice sheets (Law et al., 2020; Dunmire et al., 2020; Lampkin et al., 2020). In SW Greenland, most of the detected buried lakes had some surface meltwater within their bounds during the previous melt season (Fig. 5), supporting the hypothesis that buried lakes form due to surface lake freeze-over followed by insulating snowfall. This process is further illustrated and confirmed by a series of chronological satellite images of a buried lake detected following the 2019 melt season (Fig. 8).

In contrast, very few buried lakes in the NW and SE had any evidence of previous summer surface melt within their bounds. One possible explanation for this is that these buried lakes were once surface lakes, but have since been buried for many years. For example, in the NW, several of the buried lakes detected following the 2018 and 2019 melt seasons last appeared on the surface during the 2012 melt season, which was another exceptionally high melt year (Nghiem et al. (2012). Fig. 9a). Lampkin et al. (2020) show that buried lakes which exist for multiple years are larger than buried lakes which only exist for a single season.

In SE Greenland, many of the buried lakes that were detected never showed evidence of surface meltwater within their bounds in available optical imagery dating back to, and including, 2012 (i.e. Fig. 9f-i). Further, some buried lakes appear in the same location following both the 2018 and 2019 melt seasons (lake B and C in Figure 9f,h), while others appear in one year but not the other (e.g., lakes A and D in 9f,h). These results indicate that buried lakes in SE Greenland may form via a different process than elsewhere on the GrIS. One possible explanation for how buried lakes can form without initial surface melt water is via penetration and absorption of shortwave solar radiation beneath the ice, which is capable of creating layers of slush and liquid water beneath the surface (Leppäranta et al., 2013; MacAyeal et al., 2019). Additionally, many of these buried lakes are




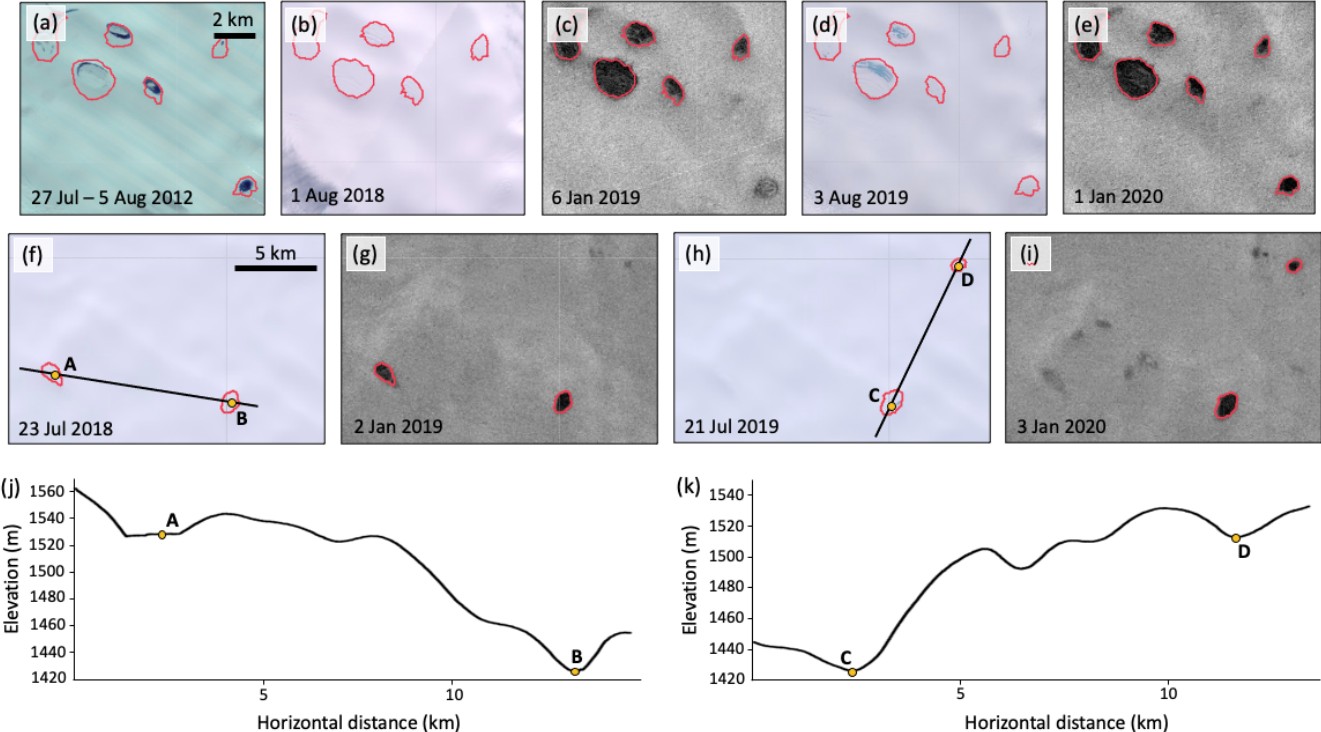

**Figure 9.** Optical and microwave images of buried lakes in NW Greenland (a-e) and SE Greenland (f-i) in chronological order. (a) Landsat 7 composite image from 27 July 27 - 5 August 2012. Red outlines indicate the buried lakes that were detected following the 2019 melt season. (b) S2 image from 1 August 2018 with post-2018 melt season buried lakes outlined in red. (c) S1 image from 6 January 2019. (d) S2 image from 3 August 2019 with post-2019 melt season buried lakes outlined in red. (e) S2 image from 1 January 2020. (f) S2 image from 23 July 2018 with post-2018 melt season buried lakes outlined in red. (g) S1 image from 2 Jan 2019. (h) S2 image from 21 July 2019 with post-2019 buried lakes outlined in red. (i) S1 image from 3 January 2020. (j) Surface elevation profile from ArcticDEM Mosaic (Porter et al., 2018) over buried lakes A and B detected following the 2018 melt season. (k) Surface elevation profile from ArcticDEM Mosaic over buried lakes C and D detected following the 2019 melt season.

located directly below surface depressions (Fig. 9j,k). Subsurface layers of snow and ice often mirror the surface topography (Drews et al., 2020), meaning that meltwater can percolate through the firn and collect on top of relatively lower ice layers located below surface topological depressions, providing another possible explanation for the formation of these features.

## 4.3 Climate drivers of buried lake distribution

Using a regional climate model to analyze temperature and melt anomalies from 2018 and 2019, we show that an increase in buried lake area in 2019 in northern Greenland (NW, NO and NE regions) is likely due to a combination of anomalously high late-season melt and anomalously high near-surface autumn (i.e. August to November) temperatures. Warmer late-summer and autumn air temperatures may prevent the water in surface lakes from freezing entirely and allow some liquid meltwater



to remain buried. This analysis suggests that increasing air temperatures, which contribute to increased summer melt, will likely lead to an increase in the number and area of buried lakes, and therefore an increased volume of liquid water that is stored beneath the surface during the winter. Surface lake distribution across the GrIS has expanded inland to higher elevations (Leeson et al., 2015) and it is expected that surface lakes will continue to form at higher elevations as air temperatures continue to rise in the future (Howat et al., 2013). Thus, we also expect that buried lakes will form further inland at higher elevations in

a warming climate, with potential implications on changing surface hydrology and total ice loss.

### 4.4 Buried lake implications

Perennial buried lakes store meltwater that could otherwise runoff and eventually drain into the ocean. Thus, these features act as a potential buffer against sea level rise. However, this buffer may only be temporary given that buried lakes have also been shown to drain, even sometimes during the winter, which is perhaps a result of ice dynamics (Benedek and Willis, 2020).

At lower elevations (< 1600 m, Poinar et al. (2015)), winter drainage of buried lakes could provide an influx of water to the bedrock, thus leading to melt-induced ice acceleration during the winter months. Further, some buried lakes in the SE and NW regions of the GrIS are located directly above observed firn aquifers. Drainage of these buried lakes could therefore provide an influx of meltwater to these firn aquifers.

### 5 Conclusions

In this paper, we have developed a method using a convolutional neural network to detect buried lakes across the entire GrIS following each of the 2018 and 2019 melt seasons. We compare buried and surface lake spatial distributions across six GrIS subregions and use a regional climate model to explain the differences. We find that while the total surface lake areal extent was less in 2018 than in 2019 across all six subregions, buried lake extent is roughly equal in 2018 and 2019 in the SW and CW regions, but much less in 2018 in the NO and NE regions. These regional differences in buried lake extent between 2018 and

2019 can be explained by regional patterns of late-season (August) melt and autumn (September - November) air temperatures. Anomalously high late-season melt, coupled with anomalously high fall air temperatures in 2019 in northern Greenland likely contributed to increased buried lake extent in this region following the 2019 melt season.

We also examined the buried lake formation process by investigating S1 and S2 imagery prior to the detection of buried lake features. We find that different formation processes likely occur in different regions. In SW and CW Greenland, buried lakes

likely form when surface lakes partially freeze over become insulated by following snowfall. In contrast, in SE Greenland, no evidence of surface melt exists in optical imagery prior to the discovery of the majority of detected buried lakes, indicating that these features form via a different process. It is possible that subsurface penetration of shortwave radiation, and/or downward percolation and subsurface pooling of near-surface melt, could be responsible for the formation of buried lakes in SE Greenland.

Surface meltwater on the GrIS plays an important role in both ice sheet dynamics, as lake drainage events provide an

influx of meltwater to the bed which temporarily increases ice velocity, and direct mass loss, as meltwater runoff positively contributes to sea level rise. The evolution of surface meltwater features has been more thoroughly examined using optical





satellite imagery. In contrast, buried meltwater features are invisible in optical imagery. As a result, these features are more poorly understood than their surface counterparts. Here, we provide a method for continent-wide mapping of buried lakes and a comprehensive look at the drivers of buried lake distribution across the GrIS. To better understand the role these features play in GrIS hydrology, a longer time series of buried lake distribution is necessary.

*Code availability.* Code used for CNN model training and testing and buried lake detection can be found at https://github.com/drdunmire1417/Greenland_CNN_code

*Data availability.* CNN training, validation, and testing data along with shapefiles for all detected buried and surface lakes will be available on Zenodo as soon as the paper is accepted. Firn aquifer data is available from the Arctic Data Center at https://arcticdata.io/catalog/view/doi%3A10.18739%2FA2TM72225. All other data is freely available on Google Earth Engine.

## Appendix A: Additional Tables/Figures relating to methods

| Model name | Source | Training time | Precision | Recall | F1 | AUC |
|---|---|---|---|---|---|---|
| AlexNet | (Krizhevsky et al., 2012) | 12 min 12 sec | 0.894 | 0.967 | 0.929 | 0.986 |
| DenseNet | (Huang et al., 2017) | 22 min 2 sec | 0.893 | 0.919 | 0.906 | 0.976 |
| GoogLeNet | (Szegedy et al., 2015) | 5 min 42 sec | 0.918 | 0.856 | 0.886 | 0.972 |
| MNASNet | (Tan et al., 2019) | 5 min 47 sec | 0.817 | 0.895 | 0.854 | 0.954 |
| MobileNet | (Sandler et al., 2018) | 5 min 52 sec | 0.937 | 0.856 | 0.895 | 0.956 |
| ResNet | (He et al., 2016) | 16 min 10 sec | 0.857 | 0.947 | 0.900 | 0.979 |
| ShuffleNet | (Ma et al., 2018) | 5 min 19 sec | 0.889 | 0.344 | 0.497 | 0.691 |
| VGG | (Simonyan and Zisserman, 2015) | 21 min 29 sec | 0.892 | 0.947 | 0.919 | 0.976 |

**Table A1.** Comparison of model evaluation metrics between different CNN architectures.



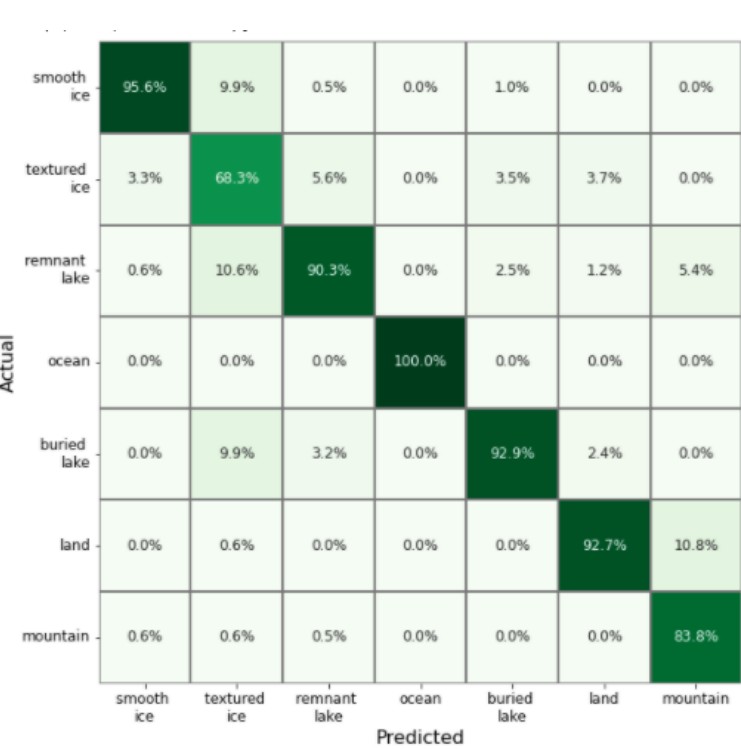

**Figure A1.** CNN test data set confusion matrix. Values are normalized by the predicted conditions, and thus represent the precision for each class. For example, a value of 92.9% for predicted = buried lake, and actual = buried lake, means that 92.9% of the predicted buried lakes are actually buried lakes.



**Figure A2.** Thresholds used in the CNN for different buried lake shapes/sizes



## Appendix B: Additional Tables/Figures relating to results

|      | GrIS Region | Num. detected lakes | Avg. lake area (km$^2$) | Total lake area (km$^2$) | Average elevation (m) |
|------|-------------|---------------------|-------------------------|--------------------------|------------------------|
| 2018 | SW          | 87                  | 0.77                    | 67.39                    | 1825                   |
|      | CW          | 89                  | 0.71                    | 63.58                    | 1557                   |
|      | NW          | 119                 | 0.55                    | 65.81                    | 1122                   |
|      | NO          | 4                   | 0.30                    | 1.20                     | 990                    |
|      | NE          | 19                  | 0.49                    | 9.24                     | 1435                   |
|      | SE          | 56                  | 0.60                    | 33.86                    | 1700                   |
| 2019 | SW          | 87                  | 0.90                    | 72.36                    | 1837                   |
|      | CW          | 77                  | 0.83                    | 63.83                    | 1654                   |
|      | NW          | 208                 | 0.61                    | 127.91                   | 1156                   |
|      | NO          | 37                  | 0.24                    | 9.04                     | 1154                   |
|      | NE          | 125                 | 0.43                    | 54.02                    | 1159                   |
|      | SE          | 72                  | 0.68                    | 48.80                    | 1718                   |

**Table B1.** Detected buried lake statistics



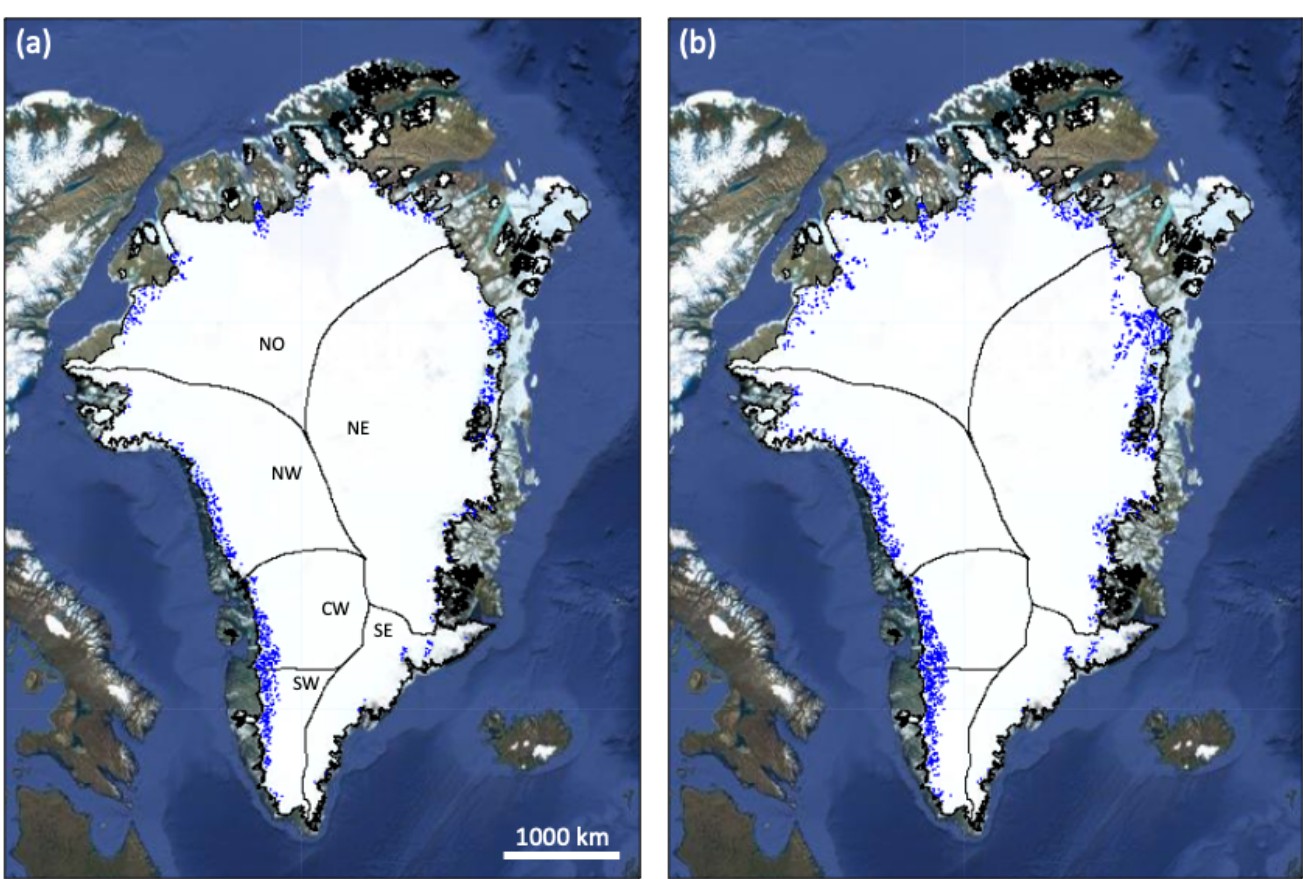

**Figure B1.** Surface lake distributions, shown in blue. (a) 2018 detected surface lakes with major GrIS drainage basins labeled (Rignot and Mouginot, 2012). (b) 2019 detected surface lakes. Background map is from GEE Gorelick et al. (2017).



|  | GrIS Region | Num. detected lakes | Avg. lake area (km$^2$) | Total lake area (km$^2$) | Average elevation (m) |
|---|---|---|---|---|---|
| 2018 | SW | 1077 | 0.34 | 367.57 | 1336 |
|  | CW | 679 | 0.45 | 304.53 | 1094 |
|  | NW | 633 | 0.24 | 154.99 | 791 |
|  | NO | 539 | 0.29 | 158.09 | 609 |
|  | NE | 705 | 0.29 | 201.72 | 718 |
|  | SE | 213 | 0.26 | 55.22 | 1289 |
| 2019 | SW | 1524 | 0.44 | 667.79 | 1386 |
|  | CW | 1000 | 0.57 | 568.93 | 1173 |
|  | NW | 1037 | 0.34 | 348.40 | 902 |
|  | NO | 754 | 0.35 | 261.55 | 796 |
|  | NE | 1540 | .42 | 640.35 | 1074 |
|  | SE | 291 | 0.28 | 81.86 | 1369 |

**Table B2.** Detected surface lake statistics



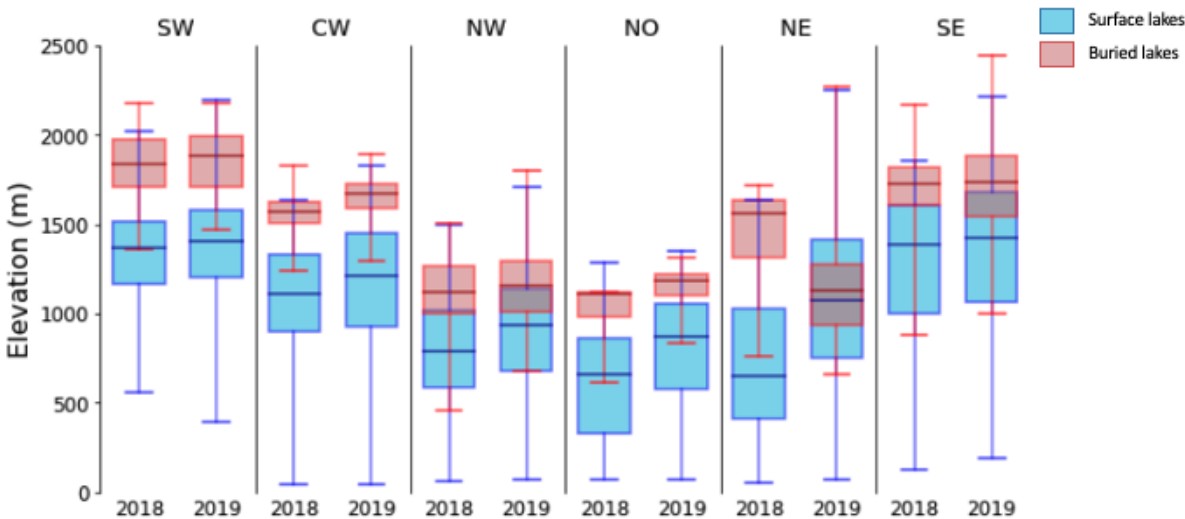

**Figure B2.** Box plots showing the distribution of detected buried (red) and surface (blue) lake elevations in each of the six GrIS subregions. Boxes represent the interquartile range of lake elevations and whiskers represent the entire range of lake elevation in a given region/year.

| | Temp 2018 | | | | | | Temp 2019 | | | | | |
| --- | --- | --- | --- | --- | --- | --- | --- | --- | --- | --- | --- | --- |
| | June | July | Aug | Sept | Oct | Nov | June | July | Aug | Sept | Oct | Nov |
| SW | 0.494 | 0.173 | -0.005 | -0.684 | -3.33 | 0.736 | 1.576 | 0.596 | -0.37 | 0.198 | 0.937 | 2.175 |
| SE | 0.157 | 0.019 | -0.06 | -0.324 | -1.227 | 0.853 | 1.179 | 0.524 | -0.016 | 0.451 | 0.291 | 1.059 |
| CW | 0.712 | -0.115 | -0.172 | -0.407 | -4.636 | 2.253 | 1.76 | 1.525 | -0.722 | 0.298 | 1.055 | 4.335 |
| NW | -0.007 | -1.368 | 1.098 | -0.488 | -3.026 | 0.557 | 0.696 | 1.48 | 0.759 | 1.605 | 3.584 | 5.033 |
| NO | -0.417 | -0.846 | 1.441 | -0.386 | -0.074 | 0.508 | 0.679 | 1.169 | 2.003 | 1.986 | 3.251 | 2.805 |
| NE | -0.261 | -0.723 | 0.417 | -0.91 | 1.352 | 1.64 | 1.19 | 1.15 | 2.001 | 2.475 | 2.244 | 2.341 |

**Figure B3.** June through November monthly temperature anomalies (°C) from the 1958-2017 mean climatology for each subregion of the GrIS in 2018 and 2019.

| | Melt 2018 | | | Melt 2019 | | |
| --- | --- | --- | --- | --- | --- | --- |
| | June | July | Aug | June | July | Aug |
| SW | -0.094 | -0.068 | 0.086 | 1.145 | 0.73 | 0.311 |
| SE | -0.086 | 0.059 | 0.158 | 0.877 | 0.503 | 0.531 |
| CW | -0.047 | -0.361 | 0.131 | 1.06 | 1.477 | 1.619 |
| NW | -0.219 | -0.663 | 0.334 | 0.614 | 0.844 | 2.705 |
| NO | -0.331 | -0.4 | 0.531 | 0.877 | 0.457 | 3.449 |
| NE | -0.232 | -0.415 | 0.227 | 1.299 | 0.799 | 3.788 |

**Figure B4.** June through November monthly melt anomaly from the 1958-2017 mean climatology for each subregion of the GrIS in 2018 and 2019.



*Author contributions.* DD conceived the study, trained the CNN, collected the necessary data to detect buried and surface lakes, and analysed buried and surface lake distributions. All authors discussed the results and were involved in editing of the manuscript.

*Competing interests.* The authors declare that they have no conflict of interest.

*Acknowledgements.* DD was supported by a NASA FINESST Fellowship (award number 80NSSC19K1329). AFB received support from the U.S. National Science Foundation (NSF) under award 1841607 to the University of Colorado Boulder. RTD was funded by the NASA ICESat-2 Project Science office. We thank Brice Noël (IMAU, Utrecht University) for providing the RACMO2 output. The ArcticDEM data was downloaded from Google Earth Engine through the Polar Geospatial Center, University of Minnesota. DEMs were created from
DigitalGlobe, Inc., imagery and funded under NSF awards 1043681, 1559691, and 1542736.



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
