# Peer review of "Contrasting regional variability of buried meltwater extent over two years across the Greenland Ice Sheet"

_The Cryosphere, 2021_

## Author Comment (AC1)

**Reviewer 1**
GENERAL
This paper presents a deep-learning application for the automated detection of buried lakes over the Greenland Ice Sheet. Subsequently, the statistics and regional differences in buried-lake presence are analyzed to infer different physical processes behind the formation of these buried lakes. The manuscript is already in really good shape, both scientifically, methodologically, and in terms of language and presentation quality. It reads like a breeze.

We thank the reviewer for their positive and encouraging comments about our paper.

I would like to suggest two points of further improvement to the paper.

(1) The temperature and melt history prior to the buried lake detection is now presented in figure 7 and tables B3 and B4. However, it would be really nice and more direct to include, for example, simulations of subsurface temperature from RACMO2, a simplified firn model, a very simple thermodynamical model, or from observations of subsurface temperature close to a buried lake (if these exist) to corroborate the link between climate and lake survival in fall. The present analysis isn't wrong but it is somewhat circumstantial.

We thank the reviewer for their suggestion to analyze subsurface temperatures in buried lake regions, and we have done this using SNOWPACK, which is a 1-dimensional, multi-layer snow model forced with RACMO climate data at three different locations. We chose this model since it has a detailed description of water flow, based on capillarity and hydraulic conductivity (Richards equation, Wever et al. 2014). These results are presented in the figure below (page 2), which will replace Figure 7 in the manuscript. These model simulations support our hypothesis that in the relatively warm 2019 in northern Greenland, snow layers with high liquid water content can remain liquid until the buried lake detection. To summarize the results of the figure, we will update the manuscript to include the following information:

*Higher air temperatures in each region during June and July 2019 contribute to higher ice-sheet-wide July 2019 subsurface temperatures (Fig. 7b). For sites X, Y, and Z, respectively, the average subsurface temperature in the top 7 m of the snow column is 2.06, 1.97, and 0.34 ℃ greater in July 2019 than in July 2018.*

*Further, higher air temperatures in NW, NO, and NE Greenland from August - November, 2019 lead to correspondingly higher subsurface temperatures than in 2018 in these regions (Figure 7b). For example, at Site X in Figure 7, which is located in CW Greenland, the September 2018 and 2019 temperature anomalies are -0.71 ℃ and +0.98 ℃, respectively and the average September subsurface temperature in the top 7 m of the snow column is 0.80 ℃ colder in 2019. In contrast, at Site Z in NE Greenland, the September 2018 and 2019 temperature anomalies are -2.54 ℃ and +4.04 ℃, respectively, average September subsurface temperature in the top 7 m of the snow column is 3.18 ℃ warmer in 2019. Additionally, at Site Y located in NW Greenland, Figure*

*7c shows that meltwater exists in the subsurface during both the 2018 and 2019 melt seasons, freezing through entirely in 2018, but lasting through the end of the year in 2019.*

[Figure]

**Figure 7.** Atmospheric and subsurface temperature modeling results. (a) July through October RACMO monthly temperature anomalies (℃) from 2018 (upper) and 2019 (lower) from left to right. Areas with monthly temperatures that are significantly different from the

1958-2017 climatological mean at the 95% confidence level are shaded with a cross-hatch pattern. (b) Simulated subsurface firn temperature profiles at 3 sites (X, Y, Z, locations indicated in panel (a)) from July through October for 2018 and 2019. (c) Time series of simulated liquid water content (%) with depth at site Z. Areas with >90% ice are shaded in grey.

(2) I think section 4.2 could be written even more clearly, by really separating the regimes in the SW, NW and SE even more rigorously.

We thank the reviewer for this suggestion, however it is hard to rigorously separate the regimes because likely a combination of buried lake formation processes occur in these regions. However, we will rework this section to more rigorously separate the different formation processes and explain which process may be dominant in different regions. We will also add a paragraph explaining that a combination of these formation processes likely occur in each region and even sometimes for individual lakes. The text that we will add, along with an additional figure, are below:

*"While Figure 5 indicates that the dominant buried lake formation process is different in different regions of the GrIS, likely, a combination of different formation mechanisms exist in each region. Further, in some cases, it appears that a combination of formation processes even exist for individual lakes. For example, Figure 10 includes a series of images of a buried lake detected in CW Greenland. S1 and S2 imagery show that the areal extent of the buried lake is greater than the surface lake detected during the previous melt season. These images therefore suggest that the formation of this buried lake resulted not only from the burial of a surface lake by snowfall, but also due subsurface melting and/or the downward percolation of surface meltwater."*

[Figure]

**Figure 10.** Optical and microwave images of a buried lake in CW Greenland in chronological order. (a) S1 image from 2 Jan 2019. (b) S2 image from 2 August 2019, (c) S1 image from 3 January 2019. For b) and c), the detected surface lake is outlined in blue and post-2019 buried lake is outlined in red.

Also, what is the role of burial rate (i.e. snowfall rates) in each of these areas on buried-lake formation and detection? And what is the role of the near-surface density (porosity) in sustaining lakes in each of these areas?

In section 3.2 we state: "We find no spatial or temporal patterns associated with monthly precipitation or surface mass balance anomalies that may explain surface and buried lake differences so this analysis is not included here." However, we had previously only looked into the effect of the burial rate on buried lake *distribution*. While burial rate does not appear to impact the number of buried lakes that form in a given region, we have since conducted additional analysis that indicates that it does have an impact on the buried lake formation process. We will include this additional analysis in the revised manuscript by including the following text and updated Figures 5 and B5 (also shown below).

*"We hypothesize that the dominant formation mechanism in different regions of the GrIS are linked, in part, to regional differences in annual precipitation. Generally, buried lakes that appear on the surface during the previous melt season are located in areas with relatively lower total annual precipitation (Fig. 5b). Conversely, large concentrations of buried lakes that never appear on the surface during the previous melt season are located in areas with relatively higher total annual precipitation. For example, in CW Greenland (Fig. 5c), the 1958-2017 climatological average annual precipitation that falls over the buried lakes detected in both 2018 and 2019 is 509 mm w.e/year for buried lakes which never appear on the surface during the previous melt season, and 451 mm w.e/year for buried lakes which do appear on the surface during the previous melt season. The difference in these two means is statistically significant at the 99% confidence level.*

*The observations described above therefore suggest that a mechanism by which annual precipitation may impact the buried lake formation process is through the availability of near-surface porous firn (Figure B5). In areas with relatively lower annual precipitation, there is less near-surface porous firn for meltwater to percolate, leading to increased surface ponding, and therefore potentially a greater proportion of buried lakes that form following surface pond burial relative to other regions. In contrast, in areas with relatively higher precipitation, there is more near-surface firn available to store meltwater, which may initially pool on subsurface impermeable ice layers leading to less surface ponding prior to buried lake detection."*

[Figure]

**Figure 5.** The impact of annual precipitation on buried lake formation processes. (a) Percentage of buried lakes that show any evidence of meltwater on the surface (> 5 pixels with an NDWI > 0.25) during the previous melt season, for each of the six subregions. (b) Spatial anomaly of the climatological (1958-2017) mean annual precipitation. Red dots represent buried lakes from 2018 and 2019 that never appeared on the surface during the previous melt season. Blue dots represent detected buried lakes from 2018 and 2019 that appeared on the surface during the previous melt season. (c) Climatological mean annual precipitation with buried lakes plotted in CW Greenland.

[Figure]

Figure B5. Simulated air content profiles at sites Y and Z. Firn air content (FAC) in the top 10 m and total annual precipitation from the RACMO 1958-2017 climatology for each site are noted in the table.

Finally, could you elaborate on the potential to extend the time series of buried lakes back in time, as mentioned in the last sentence? What sensors are available? How far back in time?

> To expand on this point, we will add the following sentences: *"Operation Ice Bridge radar has been used to detect buried lakes on the GrIS (Koenig et al., 2015), however due to this technique's limited spatial and temporal resolution, it is possible that some lakes could be missed, especially in regions with low spatial data coverage. Continent-wide S1 image coverage dates back to October 2014 and will be a useful tool for expanding our buried lake data set to other melt seasons in future work."*

SPECIFIC
Line 45: How can C-band and your method discriminate between buried lakes and firn aquifers?

In Greenland, firn aquifers are likely buried too deep to be detected directly by C-band radar. For example, the average depth of the perennial firn aquifer in SE Greenland is ~22 m below the surface (Miège et al., 2016); much deeper than the penetration depth of C-band radar. Thus, our method can only directly detect buried lakes, which are located at shallower depths than firn aquifers. However, temporal changes in microwave backscatter from C-band radar have been used to infer the locations of firn aquifers in Greenland. For example, on this subject of using C-band radar to detect firn aquifers on the GrIS, Brangers et al. (2020) says:

"An important hypothesis here is that the radar will likely not directly sense the water table (only for shallow perched water tables). Instead, it is likely that the slowdown in refreezing of water in the upper profile (above the water table) provides a distinct signature and serves as a proxy for the detection of the aquifers."

To clarify this point we will change (current) lines 216 to 219 to read: *"Firn aquifers, unlike buried lakes, are buried too deep to be directly detected with S1 microwave imagery. For example, the top surface of the perennial firn aquifer in SE Greenland is about 22+/-7 m below the ice surface (Miège et al., 2016) and can currently only be detected from S1 images by using the temporal change in microwave backscatter as a proxy to infer the locations of firn aquifers (Brangers et al. 2020). Although it is unclear what relationship the buried lakes and firn aquifers have, we suggest that the buried lakes may feed firn aquifers by draining vertically."*

Line 59: prior to buried lake detection: prior to the date of the buried-lake detection imagery.

We will change the sentence that contains this to: *"Finally, by investigating additional imagery prior to the date of the imagery used here for buried-lake detection, we hypothesize that a variety of processes are responsible for the formation of these features in different regions of the GrIS."*

Line 76: explain HV band

We will add *"Horizontally transmitted - vertically received (HV)"* to explain the HV band.

Line 134: was -> were

We will make this change.

Line 154: calculated -> calculating

We will make this change.

Line 211: I guess this means surface lake water presence?

We define "evidence of surface water within the bounds of a buried lake" as the presence of > 5 pixels with an NDWI > 0.25. By doing this we include slush and smaller areas of meltwater that are not necessarily classified as surface lakes during our surface lake detection. We describe this method from lines 159-160 in the current paper.

Section 4.4: given a rough estimate of buried-lake depth (1m? 5m? 10m?), is it possible to mention a range of the amount of water potentially stored in the lakes? For 1 m depth and 376 km^2 this means 0.376 Gt, for 10 m it is 3.76 Gt (please check!). On an ice sheet scale, these numbers are small. Are the effects mainly on a local scale? It would be good to elaborate here a bit more.

It would be extremely difficult to estimate the average depth of all buried lakes that we detect in our study. Benedek et al. (2021) applied the Bouguer-Lambert-Beer Law to optical imagery from the summer before the lakes became buried to estimate the depth of six buried lakes in CW Greenland. However, this method is impossible to apply to buried lakes that did not appear on the surface during the previous melt season, and also does not account for subsurface melting that may occur outside surface lake bounds, or for any subsequent freeze-through. In summary, we did not apply the method of Benedek et al. (2021) as there are too many detected buried lakes that don't appear on the surface in optical images, and without any field data of buried lake depths, there would be too much uncertainty with estimating volumes for us to include this in our paper.

We agree with the reviewer that on an ice sheet wide scale, the amount of water stored in buried lakes is relatively small. We will add the following text to the end of section 4.4: *"On an ice sheet wide scale, the amount of water stored in buried lakes is relatively small. Thus, these effects would mainly be relevant at local scales, particularly where large concentrations of buried lakes exist."*

Figures 4, 6 and 9: in the captions, please refer back to the overview map in Figure 3 for the locations of the images. This was at first unclear to me.

Thank you for pointing this out. Due to both reviewers' confusion about the locations of images, we will add small location maps of Greenland to each figure, indicating the ice sheet location of lakes shown in the figure. For Figure 6, we will also add latitude and longitude lines within the figure.

**References**

Wever, N., Fierz, C., Mitterer, C., Hirashima, H., and Lehning, M.: Solving Richard's Equation for snow improves snowpack meltwater runoff estimations in detailed multi-layer snowpack model, The Cryosphere, 8, 257-274, https://doi.org/10.5194/tc-8-257-2014, 2014

---

## Author Comment (AC2)

**Reviewer 2**

**General comments**

The Author Contributions section states that the analysis was designed, performed, and interpreted by the PhD student. She should be commended for this leadership. However, the contributions of the other authors are vague and appear not to meet the authorship criteria of "significant contribution to the research and paper preparation". The rest of my review looks beyond this problem, which probably needs addressing separate from the science revisions.

As is usually the case with papers where the first author is a PhD student, the student leads every aspect of the paper (including the study design, analysis, and actual writing etc), but the student receives guidance throughout the process in the form of (at least) biweekly meetings with their advisors/co-authors. Additionally, the advisors/co-authors help with editing the paper. All of the above was the case for this paper, which amounted to "significant contributions".

We will reword the Author Contribution Statement to read as follows: "DD conceived the study and led the training of the CNN, the data collection, and the analysis. AFB, JL and RTD helped to develop the ideas and methods throughout the study, and discussed the results. NW ran the firn model; SNOWPACK, and assisted with interpreting the simulation results. All authors involved in editing the manuscript"

The development of the CNN analysis and the generation of the lakes dataset are the center of the work and I think will be why future researchers cite this paper. These algorithms are well described (Methods and Appendix) and the choices and sensitivities are tested and quantified. Another strong aspect of this paper is the inference that buried lakes form by different mechanisms (burial versus trickle) in Western versus Southeastern Greenland. Finally, the writing is clear and easy to follow.

We thank this reviewer for positive comments about our paper.

My major criticisms relate to the cursory analysis of the RACMO data and to incomplete thinking on one of the lake formation speculations. I provide some detail on these points below.

(1) The correlation analysis of the buried lakes and the weather data was difficult to follow. This would be improved by better use of figures (see comments below regarding Figure 7) and a more formal analysis of the climate variables at the buried lakes. If I correctly understand the current analysis, the authors visually interpreted the RACMO model output (Figure 7) and their regional means (Figures B3 and B4) to draw conclusions about the differences in warmth and wetness over the two years and across different regions. This has problems because all regions have very large accumulation zone areas without lakes, where the RACMO data are therefore not meaningful. Analysis on a finer spatial scale more appropriate to the lakes, along with a more sophisticated analysis of the importance of each climate variable and each month (Figures B3-

B4) in forming a buried lake, is needed. A simple logistic regression analysis would be well suited and easy to implement.

With regards to the following reviewer's comment: "This has problems because all regions have very large accumulation zone areas without lakes, where the RACMO data are therefore not meaningful.", in our analysis of RACMO model output we were careful to only include lower elevation areas where buried lakes typically occur. In (current) lines 221-222, we clarify this with: "to investigate the discrepancies between total surface and buried lake area across the six GrIS subregions, we analyzed RACMO2 data from 2018 and 2019 **at elevations lower than 2500 m**, comparing temperature and melt with the climatological mean". We chose an elevation of 2500 m above sea level (a.s.I) as a threshold because the maximum elevation that a buried lake was detected at was 2450 m a.s.I. Thus, the temperature and melt anomaly numbers mentioned in section 3.2, as well as the numbers in Figures B3 and B4, concern only the areas of ice sheet that are below elevations of 2500 m. We apologize that this was not clear in the text and was missing from the captions in Figures B3 and B4. To make this clearer in our revised paper, we will reword areas of the text in section 3.2 and add this information to the captions of Figures B3 and B4.

This comment is also along similar lines as major comment 1 from Reviewer #1. Following their suggestion, we will include snow modeling results at 3 different buried lake locations in order to further test our hypothesized connection between climate variables from RACMO, and buried lake distribution and formation. We use SNOWPACK, a 1-dimensional, multi-layer snow model forced with RACMO climate data to investigate the link between climatological variables and subsurface conditions at our 3 locations. These results of this modeling work are presented in the figure below, which will replace Figure 7 in the manuscript. These model simulations support our hypothesis that in the relatively warm 2019 in northern Greenland, snow layers with high liquid water content can remain liquid until the buried lake detection. To summarize the results of the figure, we will update the manuscript to include the following information:

Higher air temperatures in each region during June and July 2019 contribute to higher icesheet-wide July 2019 subsurface temperatures (Fig. 7b). For sites X, Y, and Z, respectively, the average subsurface temperature in the top 7 m of the snow column is  $2.06, 1.97, and 0.34 \degree C$  greater in July 2019 than in July 2018.

Further, higher air temperatures in NW, NO, and NE Greenland from August - November, 2019 lead to correspondingly higher subsurface temperatures than in 2018 in these regions (Figure 7b). For example, at Site X in Figure 7, which is located in CW Greenland, the September 2018 and 2019 temperature anomalies are -0.71  $^{\circ}$ C and +0.98  $^{\circ}$ C, respectively and the average September subsurface temperature in the top 7 m of the snow column is 0.80  $^{\circ}$ C colder in 2019. In contrast, at Site Z in NE Greenland, the September 2018 and 2019 temperature anomalies are -2.54  $^{\circ}$ C and +4.04  $^{\circ}$ C, respectively, average September subsurface temperature in the top 7 m of the snow

column is 3.18 °C warmer in 2019. Additionally, at Site Y located in NW Greenland, Figure 7c shows that meltwater exists in the subsurface during both the 2018 and 2019 melt seasons, freezing through entirely in 2018, but lasting through the end of the year in 2019.

---

## Author Response (AR1)

**Editor Response**

We thank the editor for her helpful suggestions to improve this manuscript.

With regard to the additional Figure requested by Reviewer 2: I suggest that this is included in the Appendix.

Thank you for the suggestion. We have added the figure requested by Reviewer 2 to the appendix (Fig. B2).

I also request a number of minor amendments (listed below) in addition to those suggested by the authors:

Figure 3: can the projection now used in the revised Fig. 7 also be used in Fig. 3? We have updated the projections of all GrIS maps (including Fig. 3) to match that used in the revised Fig. 7.

I agree with the compromise text you suggest at the end of Section 4.4 ("On an ice sheet wide scale, the amount of water stored in buried lakes is relatively small. Thus, these effects would mainly be relevant at local scales, particularly where large concentrations of buried lakes exist."). However, would it also be useful to add a sentence justifying your decision not to characterize the water depth?

Actually, as it is simply not possible to calculate buried lake depths from SAR data (i.e. as we also explain in response to Reviewer 1's comment on page 9), we do not think we need to justify our decision for not calculating lake depths in this study. We feel that doing so would add unnecessary text to the manuscript.

In your amended paragraph copied below, I would suggest a brief rewrite to improve the structure by reducing the sentences beginning with superfluous conjunctive adverbs ('further' 'for example' 'additionally' 'in contrast'). The meaning will not change, but the readability may improve. "Further, higher air temperatures in NW, NO, and NE Greenland from August - November, 2019 lead to correspondingly higher subsurface temperatures than in 2018 in these regions (Figure 7b). For example, at Site X in Figure 7, which is located in CW Greenland, the September 2018 and 2019 temperature anomalies are -0.71 °C and +0.98 °C, respectively and the average September subsurface temperature in the top 7 m of the snow column is 0.80 °C colder in 2019. In contrast, at Site Z in NE Greenland, the September 2018 and 2019 temperature anomalies are -2.54 °C and +4.04 °C, respectively, average September subsurface temperature in the top 7 m of the snow column is 3.18 °C warmer in 2019. Additionally, at Site Y located in NW Greenland, Figure..."

We have updated the text in this paragraph by reducing the number of conjunctive adverbs used as the editor has suggested.

**Reviewer 1 Response**

**GENERAL**

This paper presents a deep-learning application for the automated detection of buried lakes over the Greenland Ice Sheet. Subsequently, the statistics and regional differences in buried-lake presence are analyzed to infer different physical processes behind the formation of these buried lakes. The manuscript is already in really good shape, both scientifically, methodologically, and in terms of language and presentation quality. It reads like a breeze.

**We thank the reviewer for their positive and encouraging comments about our paper.**

I would like to suggest two points of further improvement to the paper.

(1) The temperature and melt history prior to the buried lake detection is now presented in figure 7 and tables B3 and B4. However, it would be really nice and more direct to include, for example, simulations of subsurface temperature from RACMO2, a simplified firn model, a very simple thermodynamical model, or from observations of subsurface temperature close to a buried lake (if these exist) to corroborate the link between climate and lake survival in fall. The present analysis isn't wrong but it is somewhat circumstantial.

We thank the reviewer for their suggestion to analyze subsurface temperatures in buried lake regions, and we have done this using SNOWPACK, which is a 1-dimensional, multilayer snow model forced with RACMO climate data at three different locations. We chose this model since it has a detailed description of water flow, based on capillarity and hydraulic conductivity (Richards equation, Wever et al. 2014). These results are presented in the figure below (page 3), which will replace Figure 7 in the manuscript. These model simulations support our hypothesis that in the relatively warm 2019 in northern Greenland, snow layers with high liquid water content can remain liquid until the buried lake detection. To summarize the results of the figure, we will update the manuscript to include the following information:

Higher air temperatures in each region during June and July 2019 contribute to higher icesheet-wide July 2019 subsurface temperatures (Fig. 7b). For sites X, Y, and Z, respectively, the average subsurface temperature in the top 7 m of the snow column is 2.06, 1.97, and 0.34  $\degree$  greater in July 2019 than in July 2018.

Further, higher air temperatures in NW, NO, and NE Greenland from August - November, 2019 lead to correspondingly higher subsurface temperatures than in 2018 in these regions (Figure 7b). For example, at Site X in Figure 7, which is located in CW Greenland, the September 2018 and 2019 temperature anomalies are -0.71  $^{\circ}$  and +0.98  $^{\circ}$ C, respectively and the average September subsurface temperature in the top 7 m of the snow column is 0.80  $^{\circ}$ C colder in 2019. In contrast, at Site Z in NE Greenland, the September 2018 and 2019 temperature anomalies are -2.54  $^{\circ}$ C and +4.04  $^{\circ}$ C, respectively, average September subsurface temperature in the top 7 m of the snow column is 3.18  $^{\circ}$ C warmer in 2019. Additionally, at Site Y located in NW Greenland, Figure 7c shows that meltwater exists in the subsurface during both the 2018 and 2019 melt seasons, freezing through entirely in 2018, but lasting through the end of the year in 2019.

---

## Editor Decision (ED1)

Dear Ms. Dunmire and co-authors,

Thank you for the submission of your response to the Reviewers' comments. I am pleased to see a positive set of reviews, with some constructive suggestions that you have addressed in your initial response. The revised Figure 7 is a significant improvement.

I am content that the Revised Author Contribution Statement addresses the Reviewer's concerns, and adequately represents the workload division. However, it does not yet state who *wrote* the manuscript – please include this!

With regard to the additional Figure requested by Reviewer 2: I suggest that this is included in the Appendix.

I now request that you upload your updated manuscript including the changes detailed in your response. I also request a number of minor amendments (listed below) in addition to those suggested by the authors:

Figure 3: can the projection now used in the revised Fig. 7 also be used in Fig. 3?

I agree with the compromise text you suggest at the end of Section 4.4 (*"On an ice sheet wide scale, the amount of water stored in buried lakes is relatively small. Thus, these effects would mainly be relevant at local scales, particularly where large concentrations of buried lakes exist.").* However, would it also be useful to add a sentence justifying your decision not to characterise the water depth?

In your amended paragraph copied below, I would suggest a brief rewrite to improve the structure by reducing the sentences beginning with superfluous conjunctive adverbs ('further' 'for example' 'additionally' 'in contrast'). The meaning will not change, but the readability may improve.

*Further, higher air temperatures in NW, NO, and NE Greenland from August - November, 2019 lead to correspondingly higher subsurface temperatures than in 2018 in these regions (Figure 7b). For example, at Site X in Figure 7, which is located in CW Greenland, the September 2018 and 2019 temperature anomalies are -0.71 °C and +0.98 °C, respectively and the average September subsurface temperature in the top 7 m of the snow column is 0.80 °C colder in 2019. In contrast, at Site Z in NE Greenland, the September 2018 and 2019 temperature anomalies are -2.54 °C and +4.04 °C, respectively, average September subsurface temperature in the top 7 m of the snow column is 3.18 °C warmer in 2019. Additionally, at Site Y located in NW Greenland, Figure…*

Thank you for your contribution to The Cryosphere, I look forward to reading your revised manuscript.

Kind regards,

Dr Liz Bagshaw